# Enhanced Electrocatalytic Detection of Choline Based on CNTs and Metal Oxide Nanomaterials

**DOI:** 10.3390/molecules26216512

**Published:** 2021-10-28

**Authors:** Gloria E. Uwaya, Omolola E. Fayemi

**Affiliations:** 1Department of Chemistry, Faculty of Natural and Agricultural Sciences, North-West University (Mafikeng Campus), Private Bag X2046, Mmabatho 2735, South Africa; gloriauwaya6@gmail.com; 2Material Science Innovation and Modelling (MaSIM) Research Focus Area, Faculty of Natural and Agricultural Sciences, North-West University (Mafikeng Campus), Private Bag X2046, Mmabatho 2735, South Africa

**Keywords:** electrocatalytic activity, choline, electrochemical sensor, carbon nanotubes, metal oxide, nanomaterials

## Abstract

Choline is an officially established essential nutrient and precursor of the neurotransmitter acetylcholine. It is employed as a cholinergic activity marker in the early diagnosis of brain disorders such as Alzheimer’s and Parkinson’s disease. Low levels of choline in diets and biological fluids, such as blood plasma, urine, cerebrospinal and amniotic fluid, could be an indication of neurological disorder, fatty liver disease, neural tube defects and hemorrhagic kidney necrosis. Meanwhile, it is known that choline metabolism involves oxidation, which frees its methyl groups for entrance into single-C metabolism occurring in three phases: choline oxidase, betaine synthesis and transfer of methyl groups to homocysteine. Electrocatalytic detection of choline is of physiological and pathological significance because choline is involved in the physiological processes in the mammalian central and peripheral nervous systems and thus requires a more reliable assay for its determination in biological, food and pharmaceutical samples. Despite the use of several methods for choline determination, the superior sensitivity, high selectivity and fast analysis response time of bioanalytical-based sensors invariably have a comparative advantage over conventional analytical techniques. This review focuses on the electrocatalytic activity of nanomaterials, specifically carbon nanotubes (CNTs), CNT nanocomposites and metal/metal oxide-modified electrodes, towards choline detection using electrochemical sensors (enzyme and non-enzyme based), and various electrochemical techniques. From the survey, the electrochemical performance of the choline sensors investigated, in terms of sensitivity, selectivity and stability, is ascribed to the presence of these nanomaterials.

## 1. Introduction

Choline is a trimethyl-β-hydroxyethyl-ammonium (quaternary) compound found in plant and animal cells as phospholipids, phosphatidylcholine, phosphocholine, lysophosphatidylcholine, choline plasmalogens and sphingomyelin [1,2,3], with the structural formula presented in Figure 1.

Choline can be oxidized to betaine through an enzymatic process involving choline dehydrogenase and betaine dehydrogenase; hydrolyzed to trimethylamine by the bacterial deaminase enzyme; acetylated to acetylcholine by a cytosolic enzyme called choline acetyltransferase; and phosphorylated to phosphocholine by choline kinase [4,5]. Choline metabolism is closely related to that of different B vitamins and methionine. The pathways of choline and one-carbon metabolism intersect at the formation of methionine from homocysteine [6,7]. Methionine is regenerated (re-methylated) from homocysteine in a reaction catalyzed by betaine homocysteine methyl transferase, in which betaine, a metabolite of choline, serves as the methyl donor [5,8,9,10]. Choline is an essential nutrient officially established by the Institute of Medicine and is required for several physiological functions [6,11]. Figure 2 presents choline and folate metabolism.

### 1.1. Importance of Choline in the Body

Choline produces lecithin, sphingomyelin and all essential components of the cell membrane. Lecithin is a phosphatidylcholine-rich fraction synthesized during commercial purification of phospholipids and sometimes added to foods as an emulsifying agent [10]. Choline is involved in several physiological functions at different stages of the life cycle [6,13]. Choline modulates the integrity of deoxyribonucleic acid (DNA) and is required for neural coordination in the central nervous system (CNS), being a precursor to acetylcholine (ACh). ACh aids the signaling of the cell membrane, transport of lipoproteins, spinal cord structure, cognitive functioning, muscle control and metabolism of methyl groups, particularly homocysteine [1,7,8,10,11,14,15,16,17,18,19,20,21,22,23,24]. At a periconceptional stage, choline prevents neural tube defects (congenital disabilities of the brain, spinal cord or spine). At the pregnancy/prenatal period, choline aids proper brain and memory formation and maternal placental and liver function and protects the fetus from environmental abuse such as alcohol that could lead to abnormalities in behavior, organ structures, fetal loss and congenital disability [6,7,13]. Additionally, in premenopausal women and men, choline prevents subclinical organ dysfunction such as fatty liver, and liver and muscle damage [6,7]. Choline regulates the gall bladder and cholesterol metabolism and prevents excessive fat build-up in the liver [6,7,8]. Choline has been used to mitigate the effects of Parkinsonism and tardive dyskinesia [7,8,20,21,22,23]. Figure 3 presents the devastating impact of choline deficiency.

### 1.2. Sources of Choline

Choline can be obtained through de novo biosynthesis, diets and supplements which involve methylation of phosphatidylethanolamine to phosphatidylcholine [1,10,20].

#### 1.2.1. De Novo Biosynthesis of Choline

Choline can be acquired through endogenous biosynthesis in the liver by means of the phosphatidylethanolamine N-methyl transferase (PEMT) pathway [13,25]. Choline enters the mitochondrion via a specific carrier-mediated transport mechanism but is metabolized into betaine through a series of reactions, catalyzed by enzymes choline dehydrogenase and betaine aldehyde dehydrogenase [26]. Its precursors, which are exogenous agents consisting of lecithin, choline alphoscerate and citicholine, are converted to choline in vivo for de novo synthesis and maintenance of cell membrane phospholipids and neurotransmitters such as acetylcholine and dopamine [27]. Although the PEMT pathway represents an important source of choline, dietary intake of choline is necessary to maintain the normal function of cells, tissues and organs [10,13,22].

#### 1.2.2. Dietary Sources of Choline

Since de novo synthesis alone cannot adequately meet choline requirements in the human diet [8,10,13,22], there is a need for sourcing through sufficient intake from a variety of choline-rich foods (liver, fish, dairy products, whole grain, vegetables and egg, especially egg yolk), and dietary choline supplements which can be absorbed into the system by choline transporters (lipoprotein) [1,10,20,24,28,29]. A detailed list of choline-containing food is available in the United States Department of Agriculture database [29]. Adequate intake (AI) for choline can be estimated through the prevention of liver damage as assessed by measuring serum alanine [28]. Adequate choline intakes required in men, women, pregnant women, lactating women, infants and children as set by the Institute of Medicine are 425, 550, 450, 550, 125–150 and 200–325 mg/day, respectively [10,15,22,24]. However, high intake of choline results in hypotension with corroborative evidence of cholinergic side effects (sweating and diarrhea) and fishy body odor [7,9]. To prevent hypotension, the tolerable upper intake level (UL) of choline for adults and pregnant and lactating mothers is 3.5 g/day [7,9]. Choline’s health significance and effects when deficient in humans have necessitated the use of different analytical assays for its determination. However, the detection of choline is quite challenging since it lacks chromophores, fluorophores and electroactive groups [30].

### 1.3. Justification for the Review on Choline

Several analytical assays are used for the determination of choline. These include GC-MS [31,32,33], thin-layer chromatography (TLC) [28], ion chromatography [34], liquid chromatography with electrospray ionization-isotope dilution mass spectrometry (LC-ESI-IDMS) [3], high-performance liquid chromatography-fluorescence detection (HPLC-FLD) [24], high-throughput methods based on normal-phase chromatography-tandem mass spectrometry (LC-MS/MS) [20], liquid chromatography with an electrochemical detector (LC-ED) [2], paired-ion HPLC [31], proton nuclear magnetic resonance (1H NMR) [18], capillary zone electrophoresis with an indirect UV detector (CE-UV) [35], enzyme-based chemiluminescence [36,37] and flow injection analysis [1,17]. These methods have been found to be reliable and applicable for choline sensing in real samples. However, they are time demanding, costly and cumbersome with complex instrumentation which makes them not readily accessible for routine laboratory practices. These, or part of these drawbacks, have been overcome by introducing electrochemical sensors.

Electrochemical choline sensors, which could be non-enzyme or enzyme based, also known as the biosensors, employ some electrochemical methods such as cyclic voltammetry (CV), square-wave voltammetry (SWV), electrochemical impedance spectroscopy (EIS) and amperometry. The analysis response times of these sensors are relatively fast with superior sensitivity and selectivity, and at low cost in contrast to conventional methods. However, in order to enhance the performance of these sensors, nanomaterials such as carbon nanotubes (CNTs), which could be single walled (SWCNT) or multi-walled (MWCNT), and metal oxide nanomaterials are coupled with the sensors to improve their performance [38]. Examples of applied metal oxide nanomaterials in choline sensors include oxides of manganese, zirconium, zinc, iron (III) and nickel (MnO_2_, ZrO_2_, ZnO, Fe_3_O_4_ and NiO, respectively) [39,40,41,42,43,44]. There are very good reviews on choline determination [10,45]; however, electrocatalytic oxidation of choline specifically on carbon nanotubes and nanomaterials has not previously been explored.

This review is aimed at summarizing and highlighting the types of nanomaterials such as carbon nanotubes and metals/metal oxides employed as electrode modifiers in electrochemical sensors for the electrocatalytic detection of choline. The selectivity of the electrodes and their practical applicability for choline detection in real samples are also investigated.

## 2. Electrochemical Sensors for Choline Detection

### 2.1. Choline Biosensor (Enzymatic Electrochemical Sensors)

A biosensor is a self-contained integrated device that provides information of an analyte in the form of an electrical signal, with the aid of biological sensing elements (biomolecules) such as enzymes, antibodies, proteins, microbes and deoxyribonucleic acid (DNA) immobilized on the transducer (electrode or set of electrodes), which is an electrochemical method [38,46]. These biological sensing elements enhance the specificity and sensitivity of biosensors [38]. The electrical signal is proportional to the concentration of the detected analytes resulting from the interaction of the analytes with the immobilized biomolecules on the electrode [38,46]. Great attention is drawn to electrochemical biosensors, particularly the amperometric choline oxidase (ChOx) base, in which the choline concentration is determined indirectly depending on the determination of hydrogen peroxide (H_2_O_2_) from the catalytic oxidation of choline in the presence of oxygen [11,47,48], as represented in Equation (1).
(1)$$\mathrm{Choline}+O_{2}+H_{2}O\overset{{\mathrm{ChO}}_{x}}{\to }\mathrm{Betaine}+2H_{2}O_{2}$$

However, this approach demands a high over-potential for H_2_O_2_ oxidation, leading to interference since most electroactive species in real samples are electrochemically oxidized at this potential [39,47]. To reduce the over-potential and increase the signal, the use of horseradish peroxidase (HRP) [49,50] and, to a greater extent, nanomaterials has been introduced in biosensors [11].

Recently, nanomaterials such as carbon nanotubes (single-walled carbon nanotubes (SWCNTs) and multi-walled carbon nanotubes (MWCNTs)), polymer (polyaniline, chitosan, polyallylamine, polyvinyl sulphate) nanowires, graphene oxide and Prussian blue, together with metal and metal oxide nanoparticles such as gold (Au), platinum (Pt), zinc oxide (ZnO), zirconium (IV) oxide (ZrO_2_), manganese (IV) oxide (MnO_2_) and nickel oxide (NiO), have been incorporated in biosensors due to their possession of a high electrocatalytic effect, biocompatibility, fascinating electronic and optical properties, chemical/physical stability and high degree of electron transfer between biomolecules and the electrode surface [11,38,39,40,41,43,51]. These nanomaterials significantly impact the catalytic activities of the enzymes and increase the stability of the enzymes, which, in turn, results in an improved sensitivity and specificity of biosensors [11].

### 2.2. Non-Enzymatic Choline Electrochemical Sensors

Considerable attention has been drawn to electrochemical sensors in the detection of biological, environmental and pharmaceutical analytes due to their simplicity, stability, high sensitivity and selectivity with rapid responses and low costs. Electrochemical sensors consist of electrodes such as working, auxiliary and reference electrodes. The sensing or working electrode could be a glassy carbon, gold, carbon paste or platinum electrode and is the reaction point [46,52,53]. The counter or auxiliary electrode maintains the potential applied at the sensing electrode, and the reference electrode prevents drifting of the potential, thus providing a stable and precise voltage value [44,46,54]. The principle of electrochemical sensors is based on the reaction of the analytes of interest at the sensing electrode with a resulting electrical signal which corresponds to the concentration of analytes [54]. Signals of electrochemical sensors are amplified by modifying the sensing electrode surface with nanocatalysts, also known as nanomaterials (CNTs, metals, metal oxide nanoparticles, conducting polymers) [38]. This is due to their high electrocatalytic effect, tensile strength, chemical stability and high electron transfer rate between biomolecules and the electrode surface ascribed to the large surface area [38]. There has been little research conducted on choline oxidation using non-enzyme-based electrochemical sensors [42,44,55] which is probably due to the non-electroactive nature of choline. However, electrochemical sensors hold the possibility of detecting choline in real samples if integrated with ferromagnetic nanoparticles, which have been reported to possess an inherent enzymatic activity similar to that seen in natural peroxidase [30].

## 3. Nanomaterials and Their Electrocatalytic Activity towards Choline Detection

### 3.1. Application of CNTs (MWCNTs) and Their Composites for Choline Sensing

CNTs are a class of nanomaterial with wide application in the design of electrochemical sensors due to their fascinating structure, chemical stability, high surface area, excellent electrical conductivity, strong absorptivity, good biocompatibility and acceleration of the electron transport rate between biomolecules and the electrode [41,56,57,58,59,60,61]. Up to present, modified electrodes supported by CNTs, specifically MWCNTs and their composites (MWCNT-metal, MWCNT-metal oxide and MWCNT-polymer composites), have been applied for the electrocatalysis of choline. These nanocomposites have demonstrated improved electrocatalytic activity towards choline, which is ascribed to the stabilization of the nanoparticles (metal and metal oxide) and their sustained integrity, aided by MWCNTs, a carrier [58,62]. In addition, the synergistic catalytic effect of the nanocomposites improves the performance of choline sensors.

Qu et al. developed a biosensor using a layer-by-layer method where MWCNT and polyaniline (PANI) multilayer films were alternately assembled on a glassy carbon electrode (GCE) (MWCNT/PANI)_3_/PANI)_3_) [63]. The biosensor was thereafter amplified by immobilizing choline oxidase on the modified electrode, resulting in a ChOx/(MWCNT/PANI)_3_/PANI)_3_/GC electrode. The electrochemical performance of the biosensor to choline under the established experimental conditions (ChOx concentration, multilayer film and pH) was examined by subsequent addition of 0.05 mM choline into phosphate buffer solution (PBS), at a fixed +0.4 V potential, with a resultant 1 × 10^−6^ –2 × 10^−3^ M linear response and 0.997 correlation coefficients [63]. The limit of detection (LoD) was calculated to be 0.3 μM. The interference study of the biosensor response to choline in the midst of different species such as ascorbic acid (AA) and uric acid (UA) added successively in PBS showed non-interference in the choline signal by the various species [63]. The stability of the electrode was investigated by probing its response to 0.5 mM choline oxidation. After 15 days, the sensitivity of the electrode was the same as the initial electrode response but decreased to 80% of the original values after one month [63]. The improved catalytic activity of the electrode was linked to the protective effect of the PANI film in favor of increasing the amount of MWCNTs immobilized on the GCE and the electrostatic interaction between the negatively charged MWCNTs and positively charged PANI [63].

In another study, Purdir et al. fabricated a biosensor by co-immobilizing enzymes (acetylcholinesterase and choline oxidase) onto a nanocomposite of carboxylated MWCNT/zirconium oxide (ZrO_2_NPs) electrodeposited on the GCE surface (AChE/ChO/c-MWCNT/ZrO_2_NPs/GCE) as a working electrode for choline detection, as illustrated in Figure 1 [41].

The electrochemical behavior of choline on the proposed biosensor was examined using cyclic voltammetry in 0.1 mM choline, prepared in PBS (pH 7.4) at a 50 mV s^−1^ scan rate. The maximum current response for choline was observed at a +0.2 V potential. The sensitivity and LoD of the electrode were evaluated using amperometry at a +0.2 V potential [41]. The current response of the electrode increased in the choline chloride concentration range from 0.05 to 200 μM, with a resultant 0.01 μM LoD. The sensor was void of interference during choline measurement in the presence of AA, UA, dopamine (DA), lactic acid (LA), heparin, sodium, copper (III) sulphate (CuSO_4_), potassium chloride (KCl), sodium chloride (NaCl) and magnesium (II) chloride (MgCl_2_) of the same concentration (1.0 mM) [41]. Purdir et al. also investigated the storage stability of the biosensor at 4 °C over a period of 60 days and discovered 50% loss of the electrode initial activity towards choline, which was claimed to be superior to some choline biosensors reported in the literature [41]. The reproducibility of the electrode for five consecutive measurements within a day and after 1 week yielded a 1.35% and 2.5% coefficient of variation, indicating a good reproducibility. The designed biosensor displayed good practical application for choline detection in sera of Alzheimer’s patients and healthy persons with 98 ± 0.2 and 90 ± 0.3 mean recoveries, respectively [41]. The analytical performance of the designed biosensor was observed to be greatly enhanced with the incorporation of c-MWCNT/ZrO_2_NPs [41].

In another study, Zhang et al. reported the development of a stable choline biosensor based on the synergic effect of MWCNT and zinc oxide (ZnO) nanocomposite on a pencil graphite electrode, while ChOx was electrostatically immobilized on the modified electrode (PDDA/ChOx/ZnO/MWCNT/PGE) at an applied potential (+0.6 V) [40]. PDDA was employed as a binder. The choline dynamic concentration range was from 1 to 0.8 mM, with a sensitivity of 178 μA mM^−1^ cm^−2^ using the amperometric technique. The detection limit was found to be 0.3 μM [40]. The biosensor displayed good selectivity for choline in the presence of 0.1 mM UA, acetaminophen, AA, 0.2 mM cysteine, 7 mM glucose and 10 mM serine. Choline recovery in blood plasma samples exceeded 95%. The reproducibility of the electrode towards 0.5 mM choline from six experiments returned a 2.36% relative standard deviation (RSD) [40]. Long-term stability was established by the biosensor over 90 days, with a 5.4% decrease in the original response, which was attributed to the synergetic effects of the MWCNTs/ZnONPs [40].

Qin et al. modified a platinum (Pt) electrode with positively charged polyallylamine, negatively charged MWCNTs and polyvinyl sulphate (PVS), while poly film (PVS/PAA)_3_, serving as a perm selective layer, was alternately adsorbed continuously on the modified electrode with a multilayer film of choline oxidase/polydiallyldimethylammonium chloride (ChOx/PDDA). This resulted in a ChOx/PDDA)n/(PVS/PAA)_3_/MWCNT/Pt electrode for the determination of choline [59]. The output of the current response was optimized by increasing the ChOx/PDDA layer on the electrode, step wisely, in 3.0 × 10^−4^ M choline with ChOx/PDDA)n/(PVS/PAA)_3_/MWCNT/Pt and (ChOx/PDDA)n/(PVS/PAA)_3_/Pt electrodes. An improved current response was reported with an increase in the number of layers in the electrode with MWCNTs, owing to the excellent electron transfer ability of MWCNTs [59]. With the optimal conditions selected, a calibration curve over a linear response choline concentration range from 5 × 10^−6^–1 × 10^−4^ to 5 × 10^−7^–1 × 10^−4^ M (*S*/*N* = 3) was constructed for (ChOx/PDDA)n/(PVS/PAA)_3_/Pt) and (ChOx/PDDA)n/(PVS/PAA)_3_/MWCNTs/Pt accordingly from the amperometric measurement. LoDs of 7 × 10^−7^ and 2 × 10^−7^ M were calculated for the respective electrodes [59].

From the interference study, the designed sensors were found to be selective for choline in the presence of interfering species such as AA, UA and acetaminophen (AP) of equal concentration (0.1 mM) [59]. The selectivity of the electrode was linked to polymer-(PVS/PAA)_3_ which acted as an excellent selective film. A reproducibility study was conducted on the ChOx/PDDA)n/(PVS/PAA)_3_/Pt and ChOx/PDDA)n/(PVS/PAA)_3_/MWCNTs/Pt electrodes by six successive amperometric measurements for 0.1 mM choline, which yielded a 5.4 and 3.6% RSD, suggesting less reproducibility in the former [59]. The storage stability of the electrodes was investigated at the interval of 4 days for a month. The study revealed that the ChOx/PDDA)n/(PVS/PAA)_3_/MWCNTs/Pt electrode retained 89.5% of its initial sensitivity, which was superior to ChOx/PDDA)n/(PVS/PAA)_3_/Pt, with 78.5% sensitivity after a month [59].

In a different study, Qin et al. investigated the electrochemical behavior of choline on a Pt electrode modified with a nanocomposite film of choline oxidase, MWCNTs, AuNPs (GNPs) and PDDA (MWCNT/GNP/ChOx/PDDA/Pt) using amperometry. PDDA was employed as a dispersant and binding material. The electrode displayed a higher amperometric response to choline than the electrode of sole MWCNTs and GNPs which was attributed to the synergic effect of GNPs and MWCNTs leading to an enhanced interaction between the electrode and analytes (Figure 4). Thus, the MWCNT/GNP/ChOx/PDDA/Pt electrode was used for further studies [60].

Linearity of the electrode was obtained from 0.001 to 0.5 mM with 12.97 μA/mM sensitivity for MWCNTs-GNP-ChOx-PDDA/Pt [58]. From the interference study, the electrode exhibited good selectivity and sensitivity towards choline determination in the presence of possibly interfering species (0.1 M AA, 0.1 mM AP and 0.5 mM UA) [60]. Qin et al. reported an acceptable RSD value (3.7%) for five successive measurements from a reproducibility study, while 82.5% of the electrode initial current response to choline was found after one-month storage stability assessment [60].

Wu et al. employed a biosensor constructed by modification of a platinum electrode with a MWCNT/AuNP film (MWCNT/AuNPs/Pt) for the detection of choline, taking into account the merit of MWCNTs solubilized in chitosan and silica sol containing AuNPs and choline oxidase [64]. The MWCNT/AuNPs/Pt electrode exhibited better electrocatalytic activity for choline compared with AuNPs/Pt over a 0.05–1.6 mM choline concentration range, as shown in Figure 5. This suggests an improved electron transport between the analyte and electrode resulting from the presence of MWCNTs in the biosensing interface [64]. A sensitivity of 3.56 μA/mM, a 15 μM LoD and a 0.983 regression value were found for the electrode with optimal performance from the linear plot of current vs. different choline concentrations (insert in Figure 4) [64].

The modified MWCNTs/AuNPs/Pt electrode was reproducible. The RSD value for five successive 0.4 mM choline measurements was calculated to be 4.7%. The modified electrode showed high stability after a month with a slight current drop, which was attributed to the natural properties of the sol–gel, MWCNTs and AuNPs [64].

Magar et al. developed different amperometric choline biosensors. One of the biosensors was based on immobilization of ChOx on a functionalized MWCNT-modified GCE (ChOx/MWCNT/GCE); one was based on two drops of 1 μL gold nanoparticles/MWCNT/GCE (ChOx/(GNP)_2_/MWCNT/GCE); and one was based on four drops of 1 μL of GNPs (ChOx/(GNP)_4_/MWCNT/GCE). Immobilization of ChOx was conducted by glutaraldehyde cross-linking supported on the GCE surface with the aim of detecting choline quantitatively. By optimizing the operating potential, ChOx concentration and pH of the supporting electrolyte, the sensors were able to detect choline over a certain concentration range using amperometry [11]. ChOx/(GNP)4/MWCNT/GCE returned the best limit of detection (0.6 μM) with 204 μA cm^−2^ mM^−1^ sensitivity over 3–120 μM choline concentrations. The mechanism of the ion transport and the electrode surface characteristics was probed using electrochemical impedance spectroscopy (EIS) [11]. A low charge transfer resistance (*R*_ct_) value was obtained with ChOx/(GNP)_4_/MWCNT/GCE compared with the other electrodes studied. Information regarding the stability of the electrode was found to improve with the presence of AuNPs and MWCNTs. The electrode demonstrated high selectivity to choline in the midst of interference from AA, UA, DA and AC and was found to be applicable in the sensing of choline in real samples (choline) [11]. The performance of the sensor was attributed to the synergistic effect of the MWCNTs and gold nanoparticles.

Choline detection was also carried out in another study by Sajjadi et al. on a functionalized MWCNT- and ionic liquid (1-butyl-3-methylimidazolium tetrafluoroborated)-modified electrode, immobilized with ChOx/RTIL/NH_2_-MWCNT/GCE using amperometry in PBS (0.2 M, pH 7) [56]. The amperometric response of the designed biosensor towards choline was linear from 6.9 × 10^−3^–6.7 × 10^−1^ mM with a 0.998 regression value [56]. The LoD and sensitivity value were found to be 2.7 μM and 2.59 μA/mM. The designed sensor demonstrated reasonable reproducibility, high sensitivity and long-term stability [56].

It has been reported that ferromagnetic nanoparticles possess enzymatic properties which are close to what exist in natural peroxidase [30]; thus, they hold the potential of being employed in the development of non-enzymatic electrochemical sensors for choline sensing. Based on this, Uwaya et al. designed a choline sensor using functionalized MWCNTs and biosynthesized Fe_3_O_4_ from *Callistemon viminalis* extracts (leaf = Fe_3_O_4_L and flower = Fe_3_O_4_F) supported on a GCE (GCE/fMWCNT/Fe_3_O_4_) [42]. Electrocatalysis of 2mM choline on the designed sensors (GCE/fMWCNT/Fe_3_O_4_L and GCE/fMWCNT/Fe_3_O_4_F) was conducted using CV at a 25 mV s^−1^ scan rate. The current response was enhanced at the fMWCNT/Fe_3_O_4_-modified electrodes compared with the other electrodes studied, indicating a faster rate of electron transfer. The interfacial properties of the nanocomposite-modified electrodes were studied using EIS at a fixed potential of +0.5 V within a frequency range of 100 kHz to 0.1 Hz. Smaller charge transfer resistance (*R*_ct_) values of 0.587 and 0.795 KΩ were found at GCE/fMWCNT/Fe_3_O_4_L and GCE/fMWCNT/Fe_3_O_4_F, respectively, compared with the bare electrode (349 KΩ), which correlated with the CV experiments [42]. The small *R*_ct_ values and the amplified current at the nanocomposite electrodes were attributed to the Fe_3_O_4_ and fMWCNT interaction. Figure 2 represents the proposed mechanisms of electrochemical reaction of choline on the nanocomposite-modified electrode surfaces [42].

In order to ascertain the type of electrode reaction occurring and the kinetics of the electrodes (GCE/fMWCNT/Fe_3_O_4_L and GCE/fMWCNT/Fe_3_O_4_F), scan rate studies ranging from 25 to 400 mV s^−1^ were conducted in 2 mM choline prepared in LiCl of pH 7.3 using CV on the nanocomposite-modified electrode. The peak currents were found to increase with the increase in the scan rate, the plot of the peak current vs. the square root of the scan rate was linear and the charge transfer coefficients were 0.50 and 0.51 for the respective electrodes which resembles an ideal diffusion-controlled reaction [42]. Choline was determined quantitatively using SWV, with resulting 0.83 and 0.36 μM detection limits for GCE/fMWCNT/Fe_3_O_4_L and GCE/fMWCNT/Fe_3_O_4_F, respectively. The electrodes exhibited good selectivity to choline (0.1 mM) in the presence of 100 mM AA and 0.1 mM DA using chronoamperometry and SWV [42]. In addition, the reproducibity study for six successive measurements on the electrode yielded reasonable RSD values (6.2 and 4.5%). The electrodes were successfully employed in practical sensing of choline in pharmaceuticals with satisfactory recoveries [42]. The behavior of the sensor was attributed to the improved electrocatalytic activity of the nanocomposites.

### 3.2. Application of Metal Oxide Nanoparticles for Electrocatalytic Detection of Choline

Metal oxide nanoparticles (MONPs) formed from their metal salts are significant in the field of physics, chemistry and materials science [65]. MONPs possess structural forms with an electronic structure that is capable of revealing semiconductor, metallic and insulator properties. They display distinctive properties such as optical/electrical/thermal properties owing to their excellent density, large surface area, biocompatibility and band gap [66,67,68]. These properties have led to the application of MONPs in catalysis (photocatalysis, electrocatalysis). Examples of some metal oxide nanoparticles with application in choline sensors are zinc oxide (ZnO), iron (III) oxide magnetite (Fe_3_O_4_), nickel oxide (NiO) and manganese dioxide (MnO_2_) nanoparticles [39,40,41,42,44]. Metal oxide nanoparticles are synthesized either through the chemical or the green route, as summarized in Figure 6 [69,70,71,72,73].

Yu et al. investigated the electrocatalytic performance of a nafion/choline/oxidase/manganese dioxide-modified electrode (Nafion/ChOx/MnO_2_/GCE) using cyclic voltammetry in 0.1 mM choline chloride prepared in PBS pH 8 [39]. Their result showed a high redox response at the modified electrode compared to the bare electrode (Figure 7), suggesting increased activity linked to the large surface area and electrocatalytic property of MnO_2_ with high permeability [39]. The authors also examined the influence of scan rate variation from 20 to 500 mV s^−1^ on the redox current peaks using CV and discovered a linear increase in the current peaks with a more positive shift in the potential peaks as the scan rate increased, indicating a quasi-reversible redox process. A regression was obtained from the plot of the current peaks with the square root of the scan rate, suggesting a diffusion-controlled reaction [39].

The LoD and sensitivity of the electrode were determined. The oxidation peak currents were seen to increase linearly with the successively increased addition of choline to 0.1 M PBS at an applied +0.7 V potential. A 5 μM LoD value was obtained from the linear plot of the oxidation peak current vs. the concentration using amperometry [39]. The designed sensor demonstrated excellent selectivity to choline amidst different interferents (1 mM glucose, AP, DA, serine, 0.1 mM AA and 0.5 mM UA), with good resolved signals. Good reproducibility (2.7% RSD) for five scans and storage stability (85%) over a month were noticed using CV in 0.1 mM choline [39]. Additionally, the practical feasibility of the electrode for choline proved successful in milk, milk powder and feedstuff with a good recovery range (98–107) [39]. Yu et al. affirmed that the fabricated sensor holds the potential for choline determination in the food industry, feed additives and all fields due to the experimental results.

Bai et al. developed a biosensor for choline detection through a direct and simple electrochemical deposition of a bio-composite consisting of chitosan hydrogel, choline oxidase (ChO) and MnO_2_ onto a GCE. The electrocatalytic property of the electrode in the presence and absence of 0.30 mM choline (choline chloride) prepared in 0.2 M borate buffer, pH 7.8, was investigated using cyclic voltammetry at a 100 mV scan rate (Figure 8) [43]. The cyclic voltammogram showed an enhanced current response at the ChOx/MnO_2_/Chit/GCE (Figure 8A) [43], indicating a faster rate of electron transport which is due to the high electrocatalytic property of MnO_2_.

The sensitivity and linearity of the designed sensor were examined using square-wave voltammetry with a linear choline concentration in the range of 1 × 10^−5^ to 2.1 × 10^−3^ M. Parameters used were as follows: 1.0, 0, 0.004 and 0.025 V initial, final, increment and amplitude potential with a frequency of 15 Hz at an applied potential of 0.45 V [43]. The cathodic current increased significantly with increasing choline chloride concentration (Figure 8B). The fabricated sensor exhibited good selectivity to choline chloride (0.5 mM) in the presence of 0.5 mM ascorbic acid and 0.10 mM uric acid. In addition, from the reproducibility study of the electrode, a relative standard deviation of 4.4% in response to 0.10 mM choline for seven successive measurements was found, which is quite reasonable. The biosensor was found to retain 90 and 80% of its initial response after a month and 2 months, respectively, from the storage stability study [43]. The performance of the sensor was attributed to MnO_2_.

Sattarahmady et al. fabricated, characterized and found the application of chemically synthesized nickel oxide nanoparticles as a carbon paste electrode (CPE) modifier (NiO–CPE) for electrochemical determination of choline. An increase in the peak current was noticed on the modified carbon paste electrode (MCPE) compared with the CPE (Figure 9A, B). The kinetics of the electrode towards choline oxidation was determined by studying the impact of scan rate variations (1–1000 mV s^−1^) on the choline oxidation peak current [44], as presented in Figure 9C. An increase in the peak current together with the peak potentials was observed with increasing scan rates. The coefficient of electrons transferred was determined to be 0.52 from the graph of potential peaks vs. The logarithm of scan rates. Additionally, the anodic peak currents were found to be linear to the square root of the scan rate, indicating a diffusion-controlled process [44].

The limit of detection and sensitivity of the designed sensor were investigated using amperometry. From the amperometric determination of choline at a 550 mV fixed potential, a linear calibration curve was built (Figure 9D), with the linear choline concentration response from 0.25 to 6.98 mM yielding a 60.5 mAmol^−1^ Lcm^−2^ sensitivity value [44]. LoD and LoQ values of 25.4 μM and 84.7 μM were calculated. There was no observable chemical interference in the investigation of the selectivity of the designed electrode to choline in the presence of D-glucose, DA, L-AA, UA, L-Cysteine, N-acetyl-L-Cysteine, ephedrine and pseudoephedrine. RSDs of 0.73 and 4.63 were found from the reproducibility study of the modified and different electrodes by three amperometric measurements. A 4% drop in the peak current was noticed from the stability study of the electrode after 50 cycles [44].

A summary of the surveyed electrochemical sensors (enzyme and non-enzyme based) is presented in Table 1. 

## 4. Conclusions

This review describes electrochemical sensors for choline detection with fast response times based on the electrocatalytic activity of MWCNT, MWCNT composite and metal oxide nanomaterials as electrode modifiers. The literature review shows a wide use of electrochemical biosensors with immobilized biological sensing molecules and little application of non-enzyme-based sensors. The sensitivity and selectivity of the different sensors were also investigated. The chemical selectivity and rapid detection of choline amidst potential biomolecules interferents such as uric acid, lactic acid, dopamine, ascorbic acid, cysteine, glucose, acetaminophen and acetylcholine, to mention but a few, are important targets of electroanalytical research. This is because the potentials of most of these possible interfering species are quite close, which could possibly lead to potential overlap. To this end, highly selective recognition matrices such as polymers, nanomaterials and biological recognition elements such as enzymes have been incorporated into choline sensors to improve their selectivity in the presence of possible interferents. Hence, the selectivity of the fabricated choline sensors reviewed in the presence of the aforementioned possible interfering species was evaluated by various authors. The performance of the electrochemical sensors in choline detection investigated in terms of sensitivity and selectivity was ascribed to the presence of the nanomaterials. This could be due to their high electrical conductivity, biocompatibility and high surface area which enhanced the rate of electron transport between choline and the electrode surfaces. Considering the health importance and the devastating effect of choline when inadequate, new and more research on choline sensors employing different nanomaterials with application in real samples is imperative.

## Data Availability

The Data supporting this study are obtainable from Elsevier but restrictions apply to the accessibility of these data, which were used under license for the present study, hence, they are not publicly available. However, data are obtainable from the authors with consent of Elsevier.

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
