# Peer review of "Enhanced Electrocatalytic Detection of Choline Based on CNTs and Metal Oxide Nanomaterials"

_molecules, 2021, doi:10.3390/molecules26216512_

Round 1
Reviewer 1 Report
This mini review focuses on the electrochemical sensing of choline and recent advancements in the applied electrode components. Efficient and rapid EC sensors for biomolecules such as choline would be of great importance in everyday medical analysis therefore the topic is very important. I think that this mini review can be useful for those who are unfamiliar with this field, but also for those who are working in this field. I suggest the following corrections before acceptance:
- Abstract: 'Electrocatalytic oxidation of choline (state the significance of electrocatalytic oxidation of choline here).' - this sentence is unfinished, please, correct. Also, 'The study revealed...' sounds rather like a conclusion from an original research article and not a review. Please, rephrase. 'Enhanced' does not fit as keyword.
- Fig. 1: please, unify letter types (bold or plain).
- The list of physico-chemical properties of choline should be deleted. Instead, a graphical overview of choline metabolism would be helpful for the readers (following the first paragraph in page 2).
- Please fix the use of 'de novo', 'in vivo' - all should be Italic, with no hyphen.
- Page 3: 'egg yolk and also' - unfinished sentence.
- Page 3: the letter in the last paragraph is small. As for the contents: sensors hardly detect choline by using EIS.
- At the end of introduction: '...employed as electrode modifiers in electrochemical sensors towards electrocatalytic oxidation of choline.' - electrocatalytic degradation and sensing are two different aspects. Please, clarify the bases (types) of choline sensing by the modified electrodes in order to better introduce the follow-up sections.
- Give numbers to equations and cite those in the text, please.
- Some relevant references are missing, please include and discuss:
https://doi.org/10.1016/j.sbsr.2019.100302 https://doi.org/10.1039/C9RA07459G https://doi.org/10.1039/D0AN01939A - Table 1 is numbered as Table 2, please, correct.
- Conclusions: it is underlined that the use of electrochemical sensors on real samples would require further studies. In this context, I suggest to devote some sentences to the conclusions from studies, where competing serum (bio)molecules were tested. What is our current knowledge about chemical selectivity, what are the potential types of bio-molecules that are expected to disturb detection?
Author Response
|
1 |
Reviewer (I ) comments |
Authors' response |
|
1 |
Abstract: 'Electrocatalytic oxidation of choline (state the significance of electrocatalytic oxidation of choline here).' - this sentence is unfinished, please, correct. Also, 'The study revealed...' sounds rather like a conclusion from an original research article and not a review. Please, rephrase. 'Enhanced' does not fit as keyword.
|
Thanks so much for the observations and comments. Unfinished sentence has been corrected. Significance of electrocatalytic detection of choline is stated. See yellow highlights in abstract line 8 – 11. Enhanced has been removed from the key word |
|
2 |
Fig. 1: please, unify letter types (bold or plain).
|
Thanks for the observation, All letters have been unified. Please see figure 1 page 1 |
|
3 |
The list of physico-chemical properties of choline should be deleted. Instead, a graphical overview of choline metabolism would be helpful for the readers (following the first paragraph in page 2).
|
Thanks so much for your contribution, Choline physico-chemical properties are deleted, and overview of choline metabolism is added. Please see Figure 2 and the yellow highlights on page 3 line 11-20 |
|
4 |
Please fix the use of 'de novo', 'in vivo' - all should be Italic, with no hyphen.
|
Thanks so much for the contribution. All 'de novo', 'in vivo' is now in italics with no hyphen. Please see page yellow highlight on 3, line 24 and 29 |
|
5 |
Page 3: 'egg yolk and also' - unfinished sentence. |
Thanks so much for the observation. Unfinished sentence reconstructed. Please see yellow highlights on line 33 page 3 |
|
6 |
Page 3: the letter in the last paragraph is small. As for the contents: sensors hardly detect choline by using EIS.
|
Thank so much for the observation and contribution. The letter has been adjusted. Please see yellow highlights on page 4, line 28 -43 Amperometry was employed. Please see page 9, line 12-14. |
|
7 |
At the end of introduction: '...employed as electrode modifiers in electrochemical sensors towards electrocatalytic oxidation of choline.' - electrocatalytic degradation and sensing are two different aspects. Please, clarify the bases (types) of choline sensing by the modified electrodes in order to better introduce the follow-up sections |
Thanks so much for the comment. Correction effected See line 8-11, 18-20 |
|
8 |
Give numbers to equations and cite those in the text, please |
Thanks so much for the comments. Number assigned to equation with citation of those in the text. |
|
9 |
Some relevant references are missing, please include and discuss:
|
Your contribution is highly appreciated. https://doi.org/10.1016/j.sbsr.2019.100302 and https://doi.org/10.1039/D0AN01939A have been included (see ref 55 and 48 on page 5 line 44 and 7 accordingly.
https://doi.org/10.1039/C9RA07459G was not cited because, there is no trace of the specific electrochemical method used. Also, although stated as a non enzymatic method from the caption, it was mostly referred to as biosensor with no specific biological sensing element stated. In addition, it is not within the scope of nanomaterials under study |
|
10 |
Table 1 is numbered as Table 2, please, correct.
|
Thanks so much for the observation. Correction made. Please see table 1 on page 14. |
|
11 |
Conclusions: it is underlined that the use of electrochemical sensors on real samples would require further studies. In this context, I suggest to devote some sentences to the conclusions from studies, where competing serum (bio)molecules were tested. What is our current knowledge about chemical selectivity, what are the potential types of bio-molecules that are expected to disturb detection?
|
Your contribution is highly appreciated. Sentences from the studies have been devoted to the conclusions with chemical selectivity discussed. Please refer to the conclusion section on page 15. |
Reviewer 2 Report
Manuscript ID: molecules-1412775
Type of manuscript: Review
Title: Enhanced electrocatalytic oxidation of choline based on CNTs and metal oxide nanomaterials
Authors: OMOLOLA ESTHER FAYEMI *, Gloria Ebube Uwaya
This manuscript describes the electrocatalytic oxidation of choline based on CNTs and metal oxide nanomaterials.
There are some major problems in the manuscript and a careful review is needed.
- There are several places where sloppy editing is in evidence. For eg. “carbon nanotubes (CNTs) which could be single walled or multiwalled (MWCNT, SWCNT)” abbreviation was ordered incorrectly, “(CNTs, metal, metal oxide nanoparticles, conducting polymers and carbon nanotubes)” introduce every abbreviation before using it in the text. The first time you use the term, put the abbreviation in parentheses after the full term. Thereafter, you can stick to using the abbreviation.
- Ref. 31 represents the colorimetric choline sensor. But the author included under the section of “Non-enzymatic choline electrochemical sensors”. In the section explanation also not specific to non-enzymatic choline electrochemical sensors. Please check the reference 9 for better understanding.
- Lack of adequate literature work is too evident. See Parvaneh Rahimi, Yvonne Joseph, “Enzyme-based biosensors for choline analysis: A review” Trends Anal. Chem. 110 (2019) 367, and recent year literatures (2020, 2021) are missing.
- Typo mistake Table 2 should be Table 1. In the table, please check references are repeated. For eg. Reference 6, they studied different modifications and concluded the ChOx/(GNP)4/MWCNT/GCE is better and compared with other reports. But here the author included all the modifications, it is not necessary.
For all these reasons I do not recommend the publication of this review in the molecules journal in the present form.
Author Response
|
|
Reviewer’s 2 comment |
Authors’ response |
|
1 |
There are several places where sloppy editing is in evidence. For eg. “carbon nanotubes (CNTs) which could be single walled or multiwalled (MWCNT, SWCNT)” abbreviation was ordered incorrectly, “(CNTs, metal, metal oxide nanoparticles, conducting polymers and carbon nanotubes)” introduce every abbreviation before using it in the text. The first time you use the term, put the abbreviation in parentheses after the full term. Thereafter, you can stick to using the abbreviation.
|
We thank the reviewer for the observation and contributions. Abbreviation ordered incorrectly has been corrected. Please see yellow highlights on page 4, line 33 |
|
2 |
Ref. 31 represents the colorimetric choline sensor. But the author included under the section of “Non-enzymatic choline electrochemical sensors”. In the section explanation also not specific to non-enzymatic choline electrochemical sensors. Please check the reference 9 for better understanding.
|
We thank the reviewer for the contribution. Ref. 31 which is now 30 is not a non-enzymatic choline sensor. It was referenced under the section because the statement, although rephrased in yellow highlight, on page 5, line 44–46 is originally stated in the work of He-Bin [30] |
|
3 |
Lack of adequate literature work is too evident. See Parvaneh Rahimi, Yvonne Joseph, “Enzyme-based biosensors for choline analysis: A review” Trends Anal. Chem. 110 (2019) 367, and recent year literatures (2020, 2021) are missing.
|
Thanks so much sir for the comments. Other literature works together with Parvaneh have been cited. Please see Ref. 45, 48, 55 On page 4, 5 line 40, 7 and 44 accordingly |
|
4 |
Typo mistake Table 2 should be Table 1. In the table, please check references are repeated. For eg. Reference 6, they studied different modifications and concluded the ChOx/(GNP)4/MWCNT/GCE is better and compared with other reports. But here the author included all the modifications, it is not necessary.
|
Thank you for the observation and contribution. Table 2 has been corrected to Table 1. The different modifications with exception to the ChOx/(GNP)4/MWCNT/GCE have been removed from Table 1. Kindly refer to table 1 on page 14. |

Round 2
Reviewer 2 Report
Author responses are acceptable. Accept in present form